# Development of Robot Patient Lower Limbs to Reproduce the Sit-to-Stand Movement with Correct and Incorrect Applications of Transfer Skills by Nurses

Chingszu Lin [1,*], Taiki Ogata [2], Zhihang Zhong [1], Masako Kanai-Pak [3], Jukai Maeda [4], Yasuko Kitajima [4], Mitsuhiro Nakamura [4], Noriaki Kuwahara [5] and Jun Ota [6]

1 Department of Precision Engineering, Graduate School of Engineering, The University of Tokyo, Tokyo 113-8656, Japan; zhong@race.t.u-tokyo.ac.jp
2 School of Computing, Tokyo Institute of Technology, Tokyo 152-8550, Japan; ogata.t.af@m.titech.ac.jp
3 Faculty of Nursing, Kanto Gakuin University, Yokohama 236-8501, Japan; kanaipak@kanto-gakuin.ac.jp
4 Faculty of Nursing, Tokyo Ariake University of Medical and Health Sciences, Tokyo 135-0063, Japan; jukai@tau.ac.jp (J.M.); kitajima@tau.ac.jp (Y.K.); m-nakamura@tau.ac.jp (M.N.)
5 Department of Advanced Fibro-Science, Kyoto Institute of Technology, Kyoto 606-8585, Japan; nkuwahar@kit.ac.jp
6 Research into Artifacts Center for Engineering (RACE), School of Engineering, The University of Tokyo, Tokyo 113-8656, Japan; ota@race.t.u-tokyo.ac.jp
* Correspondence: lin@race.t.u-tokyo.ac.jp

**Abstract:** Recently, human patient simulators have been widely developed as substitutes for real patients with the objective of applying them as training tools in nursing education. Such simulated training is perceived as beneficial for imparting the required practical skills to students. Considering the aging world population, this study aimed to develop a robot patient for training nursing students in the sit-to-stand (STS) transfer skill, which is indispensable in caring for elderly people. To assess a student's skill, the robot patient should be able to access the skill correctness and behave according to whether the skill is correctly or incorrectly implemented. Accordingly, an STS control method was proposed to reproduce the different STS movements during correct and incorrect applications of the skill by the nurses. The lower limbs of a prototype robot were redesigned to provide an active joint with a compliant unit, which enables the measurement of external torque and flexibility of the human joint to be reproduced. An experiment was conducted with four nurse teachers, each of whom was asked to demonstrate both correct and incorrect STS transfer skills. The results of the external torque and joint torque measured in robot's lower limbs revealed that a significant difference ($p < 0.05$) between correct and incorrect skills. It also indicates the introduction of the proposed control method for the robot can satisfy the requirement of the assessment of STS skill. Among the various measurements conducted, the external torque of the hip joint exhibited the most significant difference and therefore represented the most robust measure for assessing whether the STS transfer skill was correctly applied.

**Keywords:** educational robots; humanoid robots; human-robot interaction; nursing education; nursing skills

## 1. Introduction

With the current rapid aging of the world's population [1], the competency of caregivers in assisting patients to transfer from a sitting to a standing position has become indispensable. The sit-to-stand (STS) movement is a highly frequent activity in a patient's daily routine [2], which significantly influences the quality of a patient's life. However, it is difficult to conduct clinical training with actual patients at schools because of safety and ethical concerns. Furthermore, there is a shortage of nursing teachers [3] who are qualified to simulate a patient for students to practice with; this hinders students from

receiving sufficient amounts of clinical practice and obtaining a sufficient depth of practical experience.

*1.1. Objective and Approach*

With current advances in technology, human patient simulators (HPSs) are considered a viable solution to the aforementioned problem and has become prevalent in nursing education [4]. An HPS can emulate a living patient for training purposes; therefore, this study aimed at developing a robot patient to function as an HPS and deploying it for the training of STS transfer skills.

During STS movement, the possible effects that incorrect application of STS transfer skills by nurses could have on a patient, and the specific locations of possible influence should also be addressed. Therefore, in this study we first defined the correct and common incorrect applications of STS transfer skills as the assessment specification. Second, the particular STS movements associated with the correctly and incorrectly applied STS transfer skills were extracted from previous experimental results obtained from the interaction of nursing teachers with a simulated patient [5]. On the basis of these prior studies, the external torque and joint torque of the lower limbs were determined to be the appropriate parameters for assessing the requisite skills. Furthermore, a robot patient was developed to reproduce the STS movement, and the nursing teachers were asked to apply the STS transfer skills correctly and incorrectly. Finally, the measurements obtained from the robot's lower limbs were analyzed. Measurements that revealed the most significant differences between the correct and incorrect applications of STS transfer skills were identified and discussed.

To develop a robot patient for the training of STS transfer skills, the following challenges need to be addressed. First, a method is required to reproduce the patient's STS movement while the nurse applies the skill. One approach is to determine a patient's STS movement through the observation of a simulated patient assisted by the correct and incorrect applications of the requisite skills. Based on these observations and the concept of admittance control, the external force exerted on a patient was determined as the factor dominating the direction and speed of the patient's STS movement. Second, the robot's lower limbs should be able to reproduce the flexibility of human joints and enable the measurement of the external torque that is required for controlling its STS movement. Because a rigid joint differs from a human joint, its use may cause the trainees to learn the application of the requisite skills in an inappropriate manner. The solution to this problem is to develop the joint with a compliant unit that can measure the external torque as well as reproduce the flexibility of human joints.

*1.2. Human Patient Simulator*

The first HPS, a mannequin, was introduced in the 1950s to teach physical assessments to nursing students [6]. Since then, HPSs have been developed and widely implemented for different training tasks. For example, upper limb simulators and lower limb robots [7,8] were developed to train physical therapists, and a robotic hand was designed for rehabilitation purposes [9]. Computerized simulators have been employed for trauma management training [10], dentist training [11], and administering epidural injections [12]. A patient mannequin was developed for training nurses in changing the clothes of patients [13]. Other HPSs have been developed for medical diagnosis training for prostate and cardiac examinations [14,15]. Similarly, researchers have introduced simulators to simulate swallowing difficulties [16] and facilitate the rehabilitation process. Furthermore, previous studies [17,18] have developed and employed a robot patient for transfer training from a bed to a wheelchair.

However, only a few studies have addressed the simulation of a patient's STS movement assisted by a nurse's STS transfer skill. Examples of studies that include humanoid robots and STS control methods are those by Asimo [19], Toyota Partner Robots, HRP-2P [20], and NAO [21]. In the 1970s, one study proposed a motion controller based on zero-moment point (ZMP) control for standing motion [22], whereas [23] used ground reaction force

control and ZMP control to produce the required torque. In addition, the conventional control computation methods, center of mass (COM) [24] and center of mass pressure (COP), [25] have been employed to control robots during standing and for maintaining balance. In addition, Kuniyoshi et al. [26] investigated the dynamics involved in standing up from a lying down position through the analysis of a simulation using a simplified robot model. Some studies have also employed trajectory libraries [27], velocity estimation [28], and real-time control [29]. Furthermore, machine learning has been applied to the study of the standing function of a robot [30].

Various robots and STS control methods have been proposed in literature. The robot patient described in [17,18] comprised passive and free joints in the lower limbs; thus, it could not reproduce voluntary STS movement. In addition, the effect of the nurse's skill on the patient–nurse interaction and the patient's STS movement should be taken into account when designing control methods. Accordingly, this study developed a robot patient based on the prototype robot in [18], and proposed the STS control method to reproduce the patient's movement in accordance with the nurse's skill. The lower limbs of the robot patient were redesigned to include an active and compliant joint, and the STS movement control method was introduced. According to the results obtained in the experiment, a significant difference was obtained in the external torque and joint torque of the robot's lower limbs when the nurse incorrectly applied the STS transfer skill.

### 1.3. Sit-to-Stand Transfer Skill

STS movement includes dynamic motion that requires extensive joint movement in the lower extremities and trunk, as well as posture changes that have a significant effect on the mass load on various parts of the body [31]. Therefore, patients and elderly people who are affected by muscle degeneration with concomitant weakness in the lower limbs may not be able to complete the STS movement independently. In such cases, the assistance of nurses or caregivers may be necessary. By referring to a previous study [32] and resource materials on nursing, the correct methods and common mistakes for nurses when assisting with STS transfer were determined as follows.

- Correct method: The nurse should squat down and lower their waist to prepare to help the patient with STS transfer.
- Incorrect method: The nurse does not bend their knees and lower the waist to assist the patient with the STS movement.

This correct method takes the safety of both the patient and the nurse into consideration. Patient safety is often the principal concern when moving a patient, for example, to prevent the patient from falling. Additionally, skill-related injuries to nurses, as well as their physical burden, have been taken into consideration to ensure nurses' safety. However, STS assistance by nurses is frequently executed incorrectly, according to the resource material for nurses and discussions with nursing teachers. A common error made by students and novices is lack of appropriate preparation, which can result directly in incorrect method of STS transfer and injuries for both patient and practitioner.

The remainder of this paper is structured as follows—Section 2 presents the methodology, and Section 3 describes the development of the robot patient. Section 4 describes the experiment and subsequently outlines the results. Section 5 discusses and interprets the results in detail. Finally, Section 6 concludes the study and provides comments on future research that can expand on the outcomes achieved in this study.

## 2. Requirements of the Robot Patient

### 2.1. Measured Parameters for Assessment

To assess the STS transfer skill of a nurse, the robot patient needs to indicate when the skill is applied correctly and incorrectly. According to our prior study [5], the ankle joints of the patient's lower limbs are not sufficiently robust to evaluate the skill correctness, because the ankle joints are susceptible to the difference between the heights of the nurse and the patient. Therefore, in this study, the joint torque and external torque were determined

as the evaluation index by conducting measurements on the robot patient's lower limbs, which included the hips, knees, and ankles.

### 2.1.1. Joint Torque of Lower Limbs

STS movement includes dynamic motion that requires extensive joint movement in the lower extremities and trunk, as well as posture changes associated with the load acting on the body [31]. Previous studies have focused on observing the joint torque in the lower extremities during STS movement. For example, one study focused on estimating the torque (moment) of the hips and knees with and without arm support during STS movement [33]. Another study focused on investigating the effect of using handgrips during STS transfer, where it was found that the use of handgrips caused a significant decrease in the maximum joint moments [34]. Similarly, in [35], STS movement with handrail support was examined in elderly subjects, and it was found that the use of a handrail caused a large decrease in torque. Furthermore, [36] proposed a robot suit to support standing movements in paraplegic patients, enabling the patient to stand up with lower joint torque. Therefore, joint torque appears to be an important factor in standing behavior. Accordingly, in the current study, a hypothesis that the joint torque of the lower limbs of the robot patient can indicate the correctness of the method used by the nurse in providing standing assistance was proposed.

### 2.1.2. External Torque of Lower Limbs

STS transfer skill is defined as the appropriate support given by a nurse to help a patient in moving from a sitting to a standing position. The nature of the STS movement thus depends upon the standing assistance provided by the nurse. In previous studies, nurses were observed while demonstrating standing assistance and other patient-handling skills. For example, an electromyograph (EMG) was recorded for caregivers during STS transfer assistance with two types of wheelchairs (conventional and powered) [37]. The muscle activity, force, and joint moments of the nurses were also measured, and the different approaches to the handling tasks were compared [38]. In another study, the muscle activity of experienced and novice nurses while conducting lifting tasks were compared [39], revealing a significant difference in muscle activity between the experienced and novice nurses. The joint torque of the nurses' shoulders and their lower back compression during patient-handling tasks have also been measured and investigated [40].

To measure the torque exerted by a nurse, many studies have used sensors installed directly on the nurse's body, including EMG sensors and motion capture systems. However, the purpose of the current study was to understand the effect on the patient; in other words, the aim was to discover the influence of a nurse's supporting torque on the patient, rather than on the nurse. Furthermore, sensors attached directly to the nurses could influence their performance and would be difficult to apply in training cases that included large numbers of trainees, because of the difficulty in installing and uninstalling the sensors. Accordingly, our assessment measured the external torque in the patient's lower limbs. It was also hypothesized that there would be a difference in the external torque measured in the robot joints, depending on whether the nurse had applied the skill of providing standing assistance correctly or incorrectly.

Based on the clinical experience of nurses, the minimum supporting weight of a nurse is approximately 0.5 kg on the trunk base (i.e., the waist). The distance from the robot's waistband, where a nurse is applying the hold, to the patient's hip joint is 0.4 m. Therefore, the minimum torque was set to 0.98 Nm for each hip joint. In addition, according to experience and resource materials for nursing, it is possible to cause injury to a nurse when the mass lifted by the nurse exceeds 20 kg. Therefore, the maximum value of the external torque measurement was set to 39.2 Nm. Torques exceeding this maximum value were not encouraged in the experiment.

### 2.2. Simulation of STS Movement by Robot Patient

2.2.1. Admittance Value of Physical Interaction

To use the robot patient to reproduce the STS movement, the physical interaction between the nurse and the patient should be considered. Here, the physical interaction indicates that the speed and direction of the robot patient's movement are dominated by the nurse's supporting force.

- Magnitude of Nurse's Supporting Force
  Based on the observations made during STS transfers, it was found that if the nurse exerted a larger force while supporting the patient, the patient was able to stand up more rapidly. In contrast, the patient stood up slowly when a smaller supporting force was applied by the nurse. This observation is also related to kinematics, with respect to the relation between force and acceleration. According to this relation, the magnitude of the nurse's supporting force should proportionally influence the speed of the patient's standing movement.
- Direction of Nurse's Supporting Force
  The patient's movement generally follows the direction of the supporting force applied by the nurse. For instance, if the nurse exerts a torque to bend the patient's trunk, the patient's hip joint will be flexed. If the nurse exerts a torque to extend the trunk, the patient's trunk will be extended. Accordingly, the direction of the nurse's supporting force was used as a factor in determining the direction of patient movement during STS transfer.

These observations imply that it is necessary to set the required value of the admittance in the control method of the STS movement. Admittance is the inverse of impedance, and it represents the force–velocity relationship between the external torque (or force) from the environment and the rotational velocity of the joint [41]. First, the external torque was measured from the hip joint, because this joint is close to the waistband that the nurse uses to support the patient. To reproduce the STS movement, the value of the admittance $Z$ is determined from the following equation.

$$Z_{joint} = \tau_{external} \,/\, \dot{\theta}_{joint}. \tag{1}$$

Here, $\tau_{external}$ represents the external supporting torque; and $\theta_{joint}$ is the rotational speed of the joint. Therefore, the relationship between the nurse's supporting torque $\tau_{external}$ and the motion of the patient's lower limb joint (i.e., rotational speed $\theta_{joint}$) should be observed in advance. However, as only a few studies have measured both the nurse's supporting torque and the motion of the patient's lower limb joint during STS transfer, the available data are very limited. A preliminary study examined the data obtained from prior work in [5]. According to these results, there is a huge variation in the time required to complete the STS movement with standing assistance, ranging from 4 s to 10 s. Furthermore, we investigated the range of external torques exerted by the nurses. According to clinical experience, lifting a weight greater than 20 kg by a nurse could result in injury, such as lower back pain. Therefore, 20 kg was determined as the maximum torque, and it is recommended that nurses should not exert torque exceeding this value. Further, the nurse usually grips the patient's waistband, which is approximately 40 cm away from the hip joint. Accordingly, the maximum torque that nursing teachers recommend is 39.2 Nm on each hip joint. The quickest possible STS movement was assumed to occur when the nurse adds support with a maximum torque of 39.2 Nm. Therefore, the average rotational speed of the hip and knee joints (from 90° to 0 °) is 22.5°/s, and the corresponding admittance is $Z = 1.74$ Nm $\cdot$ (°/s)$^{-1}$. The value of the admittance $Z$ can be adjusted when simulating patients with different disabilities. A severe level of disability may correspond to a greater admittance $Z$ and therefore may require a greater torque to rotate the patient's joint.

### 2.2.2. Correct and Incorrect Applications of Skills for Assisting STS Movement

Biomechanics is concerned with the human body's structure and function, as well as the mechanical motion of the components of biological systems [42]. Accordingly, the STS movement of a patient should obey the principles of human biomechanics. If the robot does not obey the principles of biomechanics during STS transfer, the nurses may learn improper methods for providing standing assistance.

According to previous studies and an experiment conducted with a simulated patient [5,43], STS movements were categorized into two types. In Figure 1a, the movement assisted with the correct STS transfer skill is depicted. At the start of the movement, the trunk (i.e., hip joint) along with the knee and ankle joints is bent. This constitutes phase I of the STS movement and is followed by extension of the hip and knee to $0°$, while the ankle joints return to $90°$. This constitutes phase II of the STS transfer. In contrast, the incorrect application of the skill is depicted in Figure 1b, where the STS movement is shown without bending of the trunk, but with extension of the hip and knee joints from $90°$ to $0°$ in both STS phases I and II.

Other standing configurations do exist for specific patients, such as quadriplegic patients who have lost all their muscle functions. However, this study aims to simulate those patients who suffer only from weakness in the lower limbs, rather than complete paralysis. Thus, other configurations of STS movement were not included in the scope of this study.

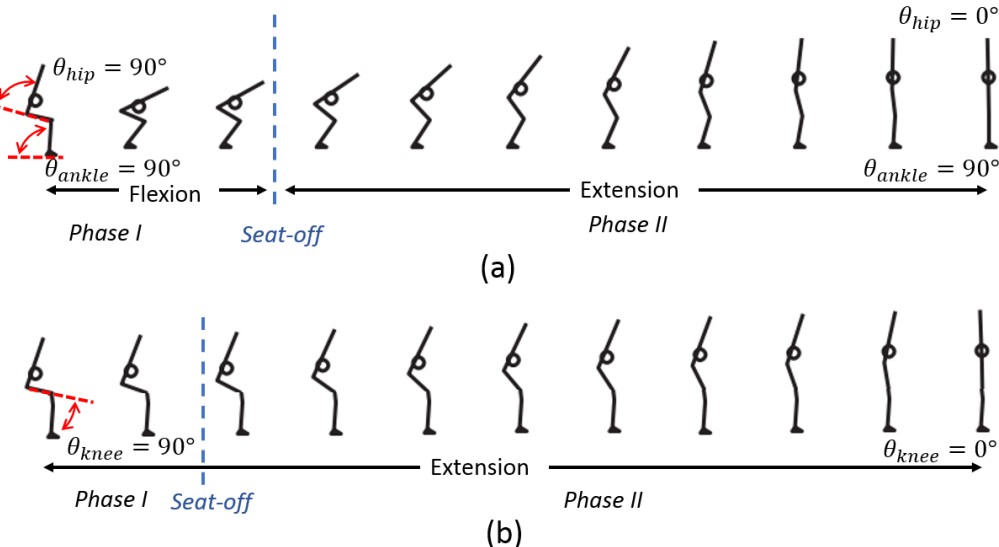

**Figure 1.** Body configuration in sit-to-stand (STS) movement (**a**) bending of the trunk followed by extending of the lower limbs by correct STS transfer skill (**b**) extending of the lower limbs without bending of the trunk by incorrect STS transfer skill.

### 2.2.3. Range of Motion and Required Torque during STS Movement

The required range of motion (ROM) of the lower limb during STS transfer was determined by referring to basic human anatomy [44] and the phases of STS movement [43] shown in Figure 1. The angle of hip joint flexion-extension without rotating the pelvis ranges from $0°$ to $120°$, whereas the angle of knee joint motion ranges from $0°$ to $150°$. Lastly, the angle of ankle joint dorsiflexion and plantar flexion ranges from $90°$ to $120°$. The robot's lower limbs were developed in accordance with these anatomical observations. The required torques were computed by analyzing the postures adopted during STS movement, within the designated ROM. The values determined through this analysis are presented in Table 1.

For the STS movement, the required torques were calculated based on the robot mass, the lengths and weights of the body segmentation, and the joint angles. The maximum required torques during the STS process occur at the joint angles indicated in Figure 2. The

mass of the upper body from the head to the hip joint is 20 kg, from the head to the knee joint is 30 kg, and from the head to the ankle joint is 34 kg. The COM is measured through the distribution of the robot's weight. Accordingly, the calculated maximum required torques are 19.6 Nm (at 120°) for each hip joint, 35.2 Nm (at 90°) for each knee joint, and 24.9 Nm (at 120°) for each ankle joint. These values represent the torques required for the robot patient to stand up without any assistance, that is, the torques required to complete the STS movement independently. Therefore, the required joint torque with the assistance of a nurse would be lower than these values.

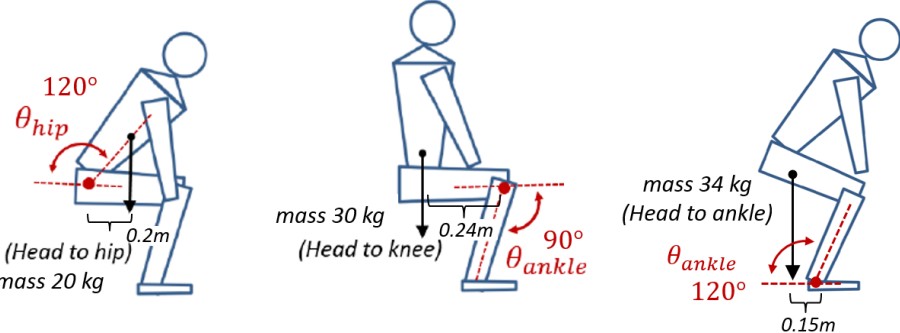

**Figure 2.** Process of STS transfer with the range of motion (ROM) and maximum angle shown.

**Table 1.** Requirements of the robot patient.

| Item | Content |
| --- | --- |
| Measurement of external torque | Range: 0.98–39.2 Nm |
| Admittance value | $Z = 1.74 \text{ Nm} \cdot (°/\text{s})^{-1}$. |
| Required torque during sit-to-stand movement | Hip joint: 19.6 Nm<br>Knee joint: 35.2 Nm<br>Ankle joint: 24.9 Nm |
| Range of movement | Hip joint: 0°–120°<br>Knee joint: 0°–150°<br>Ankle joint: 70°–120° |
| Stiffness | Hip joint: 216–266 N·m/rad |

*2.3. Stiffness of Joint*

As described in a previous subsection, the flexibility of the robot patient's trunk has to be similar to that of humans. Therefore, the required range of stiffness compliance of the hip joint was determined by referring to previous research work [45]. It has been shown that the stiffness of the human trunk ranges from 423 to 532 Nm/rad while bending during extension. To simulate the stiffness of the human trunk, the required stiffness of each robot hip joint was set to range from 216 to 266 Nm/rad.

## 3. Development of Robot Patient's Lower Limbs

*3.1. Robot Patient*

To develop a robot patient for the training of nurses in STS transfer assistance, a prototype robot's lower limbs [18] were redesigned to include active joints and an STS control method. Although the lower limbs were modified, the trunk and upper limbs remained unchanged, as shown in Figure 3. The average height of 157 cm and average mass of 51.5 kg of a Japanese female were obtained from [46]. Accordingly, the designed height of the robot was 157 cm. In addition, the statistics for the lower limb presented in [47] indicate that the average height from the ground to the waist of a Japanese female is 97.6 cm, whereas the average height to the hip of a Japanese female is 76.7 cm. The

designed mass of the robot in this study was 38 kg, which is approximately 75% of the average mass of a Japanese female. The robot's lower limb had a designed mass of 18 kg, and the upper trunk had a designed mass of 20 kg. These values were selected to avoid injuries, considering that trainee nurses might perform incorrect operations while lifting the robot patient during the transfer training.

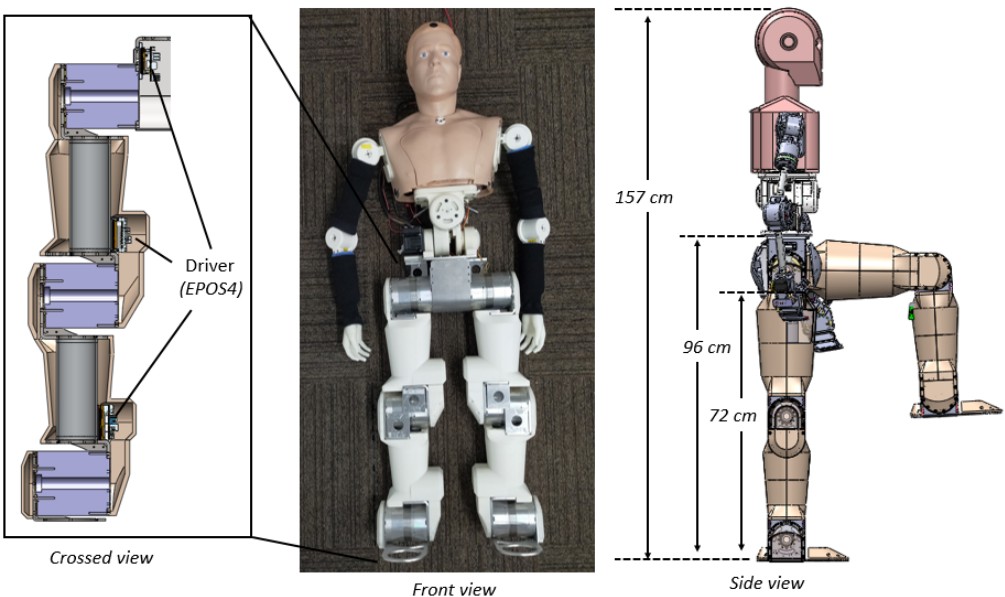

**Figure 3.** Configuration of improved robot patient with active joints in lower limbs.

### 3.2. Robot Patient's Lower Limb

In this study, a robot patient's lower limbs was developed by including compliant joints, and the specifications is presented in Table 2. The robot's lower limbs had six degrees of freedom; each of them was developed as an active joint, enabling the robot to reproduce the STS movement voluntarily. Both lower limbs comprised hip (flexion/extension), knee (flexion/extension), and ankle (flexion/extension) joints. Six modular joints of the same size were assembled to form the lower limb. Further, to measure the external torque (i.e., a nurse's supporting force), a compliant unit was developed and installed within each joint. Each motor was controlled by a driver (EPOS4, MAXON CO., LTD.), which was installed at the side joint nearby. The joint structures were made of stainless steel. The links between the joints, as well as the connection boards fixing the links to the joints, were made of aluminum and iron. Except for the main structure, the covers were fabricated with a 3D printer using acrylonitrile butadiene styrene as the material.

**Table 2.** Specifications of robot patient.

|  | Arm Joint | Waist Joint | Hip & Knee Joint | Ankle Joint |
|---|---|---|---|---|
| Torque (Nm) | 2.74 | 52 | 45.8 | 36.64 |
| Voltage (V) | 7.2 | 30 | 30 | 30 |
| Motor | Futaba RC405CB | Moog Animatics SM23165DT | Maxon EC frameless 543673 | Maxon EC frameless 543673 |
| Reduction gear | — | Harmonic driver CSD-20-100-2UP | TPI TSH-25-100-HST | TPI TSH-25-80-HST |
| Reduction ratio | — | 100:1 | 100:1 | 80:1 |

### 3.2.1. Hollow Modular Compliant Joint

The design of the lower limb joints differed from a typical joint design by being hollow, with space at the center of each joint. With the advancement in technology, hollow joints play an important role in modern robot assembly, maintenance, and wiring. The hollow shaft design enables the throughput of cables and avoids the entangling of wires. Accordingly, to achieve a hollow design, each of the joints that we developed comprised a hollow-type reduction gear, a frameless motor, and a hollow compliant unit, as shown in Figure 4. To decrease friction and ensure a degree of parallelism and concentricity, two needle bearings, one needle thrust bearing, and two deep groove ball bearings were installed in each joint. The joints were 110 mm in diameter and 112 mm in height. Additionally, to allow cables to pass through, the hollow tubes had an inner diameter of 24 mm.

### 3.2.2. Compliant Unit

Several solutions have been proposed for passive compliant joints, and the series elastic actuator (SEA) architecture, which was adopted in our design, presents a simple and effective solution [48]. This design essentially comprises a series of gear motors in a spring connected to the load [49,50]. Furthermore, the SEA design allows the force exerted on the joint to be easily evaluated by measuring the deflection of the elastic element [51].

The compliant unit for the hollow joint was designed with a stiffness of 241.3 Nm/rad, as shown in Figure 5. It comprised springs, compliant part one and part two, a small gear (pitch: 0.8, number of teeth: 16), a large gear (pitch: 0.8, number of teeth: 64), a shaft, and an angular sensor. Part one was the input and part two was the output of the compliant unit. Compliant part one was connected to the reduction gear, whereas compliant part two was fixed to the flange (the output of the joint). The springs were placed on the recesses between parts one and two. The bearings were installed between the two compliant parts and between the compliant unit and the inside of the joint, to decrease friction (see Figure 4).

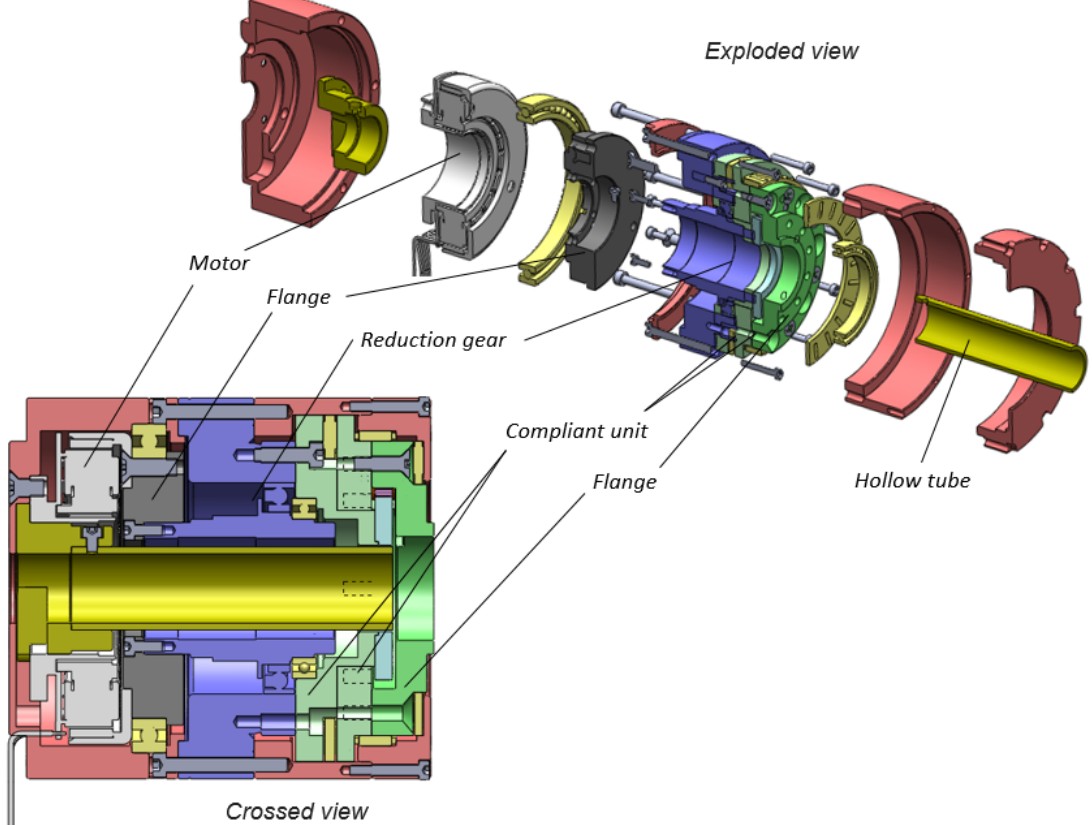

**Figure 4.** Mechanical design of hollow modular compliant joint.

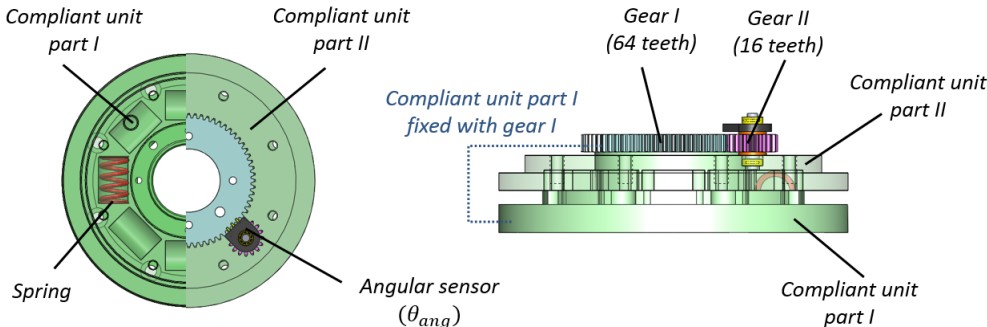

**Figure 5.** Mechanical design of a hollow compliant unit.

### 3.2.3. Measurement of External Torque

To satisfy the requirement of force sensing, the compliant unit was designed to be capable of measuring spring deformation. The relative movement between the compliant parts I and II resulted in deformation of the spring and an angular variation of the angular sensor, as shown in Figure 6. If no external force was present, the angular difference remained zero, indicating that there was no compression or deformation of the springs. If, however, an external force were applied, the springs would be compressed, causing a variation in the angular difference, which would be detected by the angle sensor of the compliant unit. The degree of angular difference allows the external torque $\tau_{external}$ to be computed using the following Equations (2)–(4).

$$\tau_{external} = \tau_{deform} \tag{2}$$

$$\Delta\theta_c = \Delta\theta_{ang} \, / \, R \tag{3}$$

$$\tau_{deform} = n \cdot \Delta S \cdot k_{compliant} = n \cdot d \cdot \Delta\theta_{ang} \cdot k_{spring} \\ \simeq n \cdot d \cdot \Delta l_c \cdot k_{spring}. \tag{4}$$

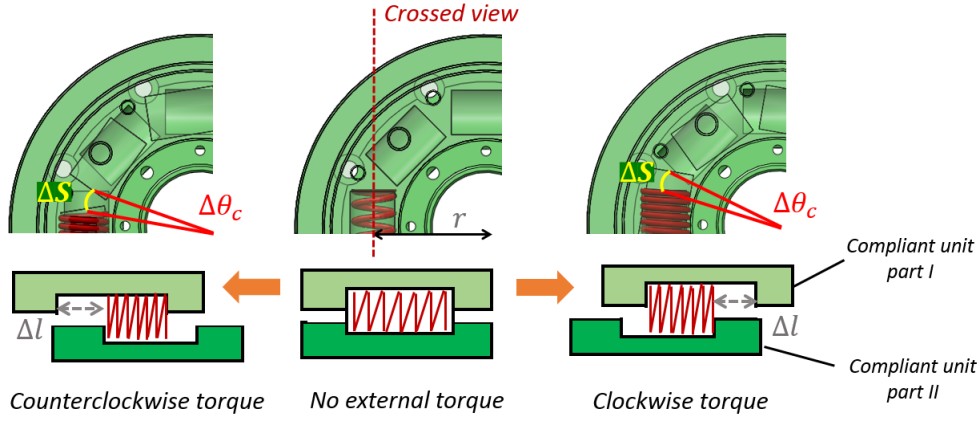

**Figure 6.** Operating principle of compliant unit when external torque is applied.

Here, $\tau_{deform}$ is the torque after the spring has been compressed; $\theta_c$ is the angular difference between compliant unit parts one and two, and $\theta_{angle}$ is the angle measured from the angular position sensor. Additionally, $R$ is the ratio of the two gears (1:4) and the ratio between $\Delta\theta_c$ and $\Delta\theta_{ang}$. The quantity $d$ is the radius (29.5 mm) from the center to the spring, and $k_{compliant}$ and $k_{spring}$ are the stiffness of the compliant unit and the elasticity coefficient of the spring, respectively. Furthermore, $n$ is the number of springs placed in the compliant unit (four in the current study). $\Delta S$ and $\Delta l_c$ are the variations in arc length and linear distance when the springs are deformed. The design of the compliant unit allows the maximum deformation of the spring when $\Delta\theta_{ang}$ equals 10 °. Therefore, the measurable torque ranges from 0 to 42 Nm.

### 3.2.4. Measurement of Joint Torque

During the experiment, both the motor current and the external force were measured in the robot's lower limb joint. The motor current generated a torque $\tau_{motor}$ in accordance with the schematic diagram of electromechanical systems. The armature was a rotating circuit through which a motor current $i_{motor}$ flowed. When the armature passed through the constant magnetic flux $\Phi$ of a permanent magnet, called the fixed field, the resulting torque $\tau_{motor}$ turned the rotor. Accordingly, the motor torque and joint torque are given by (5) and (6):

$$\tau_{motor} = K_t \cdot i_{motor} \tag{5}$$

$$\tau_{joint} = \tau_{motor} \cdot r, \tag{6}$$

where $i_{motor}$ is the armature current, $K_t$ is the torque constant of 71.3 mNm/A for the Maxon EC frameless motor, and r is the ratio of the reduction gear.

### 3.3. Control Method of STS Movement

To propose a method for controlling the STS movement of the lower limbs, the interaction between the robot's movement and the nurse's skill should be reproduced; therefore, the robot's movement must follow the external torque measured at the joint. The concept of admittance control of the compliant unit allows the dynamic relationship between the actuator velocity and the applied external forces to be shaped. This study adopted admittance control to exploit its force-sensing capability in the compliant unit and simplify the controlling mechanism. With this implementation, the inner position loop can be run rapidly, while the outer force loop is responsible for shaping the force–velocity (or force–position) relationship [52].

The STS control method was proposed as shown in Figure 7. For the postural control of the robot's lower limbs, the desired final standing position was incorporated into the planning of the motion. The joint angles ($\theta_{hip}$, $\theta_{knee}$, and $\theta_{ankle}$) were measured and incorporated into the planning of the motion as the factors for generating the motion command of the motor. In addition, the external torque ($\tau_{external.hip}$) measured from the hip joint, (which is used to evaluate whether the nurse has correctly applied the skill) was adopted as the factor for computing the motor's motion command in each joint. The motion command of the motor is generated by the motion planning and sent to the profile position mode to form the trajectory. A current valve for limiting the current passing to the motor was incorporated in the position loop. The function of current limitation was implemented to reproduce the weakness of the patient's lower limbs.

Motion planning in the postural control is presented in Figure 8. The objective of motion planning involves moving the joints from a sitting ($\theta_{hip} = 90°$, $\theta_{knee} = 90°$, $\theta_{ankle} = 90°$) to a standing position ($\Delta\theta_{hip} = 0°$, $\theta_{knee} = 0°$, $\theta_{ankle} = 90°$). The first step of motion planning is to measure the external force, and the program continues only if the external torque is not between the low ($Th_{low}$) and high thresholds ($Th_{high}$). Otherwise, it repeatedly reads the external torque. The required elements in motion planning include the external torque ($\tau_{external}$) measured at the hip joint and the absolute position of the joint angles. These values are measured and used for computing the commands that move the motor. The absolute position of each joint allows us to ensure that the STS movement proceeds as defined for the correct and incorrect applications of the skill by the nurse. The motor commands generated by the motion planning are the relative position value ($\Delta\theta$) and the angular velocity ($w$). The reading of the values and consequent generation of the motor commands is repeated until the robot reaches the final standing position. In the motion planning, $\alpha$ and $\beta$ were set as $1°$, representing the relative movement distance of the joint during a loop. The value of $r$ is the ratio (1:100) of reduction gear in the hip and knee joints, and the value $\gamma$ is the ratio of revolutions per minute (RPM) to degrees per second.

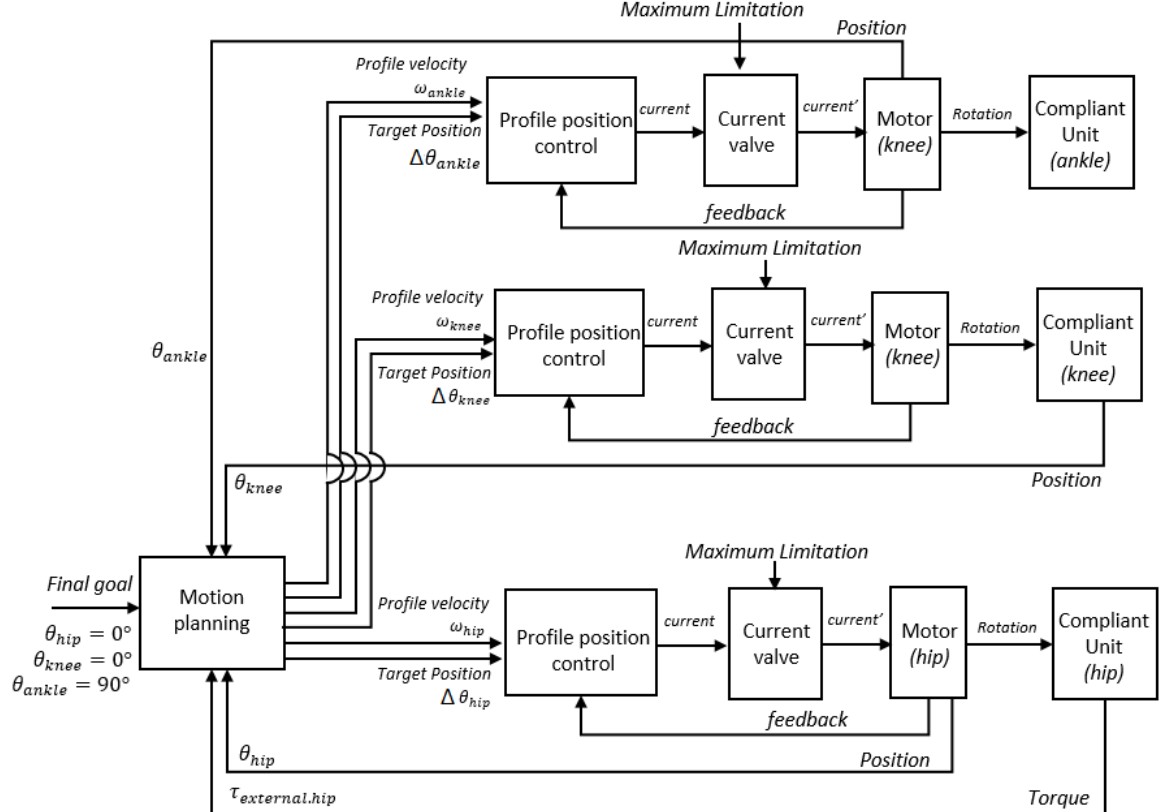

**Figure 7.** Proposed control method of STS movement.

Based on the interaction between the patient's movement and the nurse's skill during the STS transfer, the patient's standing speed depends on the quantity of external torque (i.e., the nurse's supporting force), which is determined by the admittance value $Z$ (1.74 Nm $\cdot$ ($^\circ$/s)$^{-1}$). Furthermore, the patient's posture moves according to the direction of the external torque. This direction is represented by either a positive or a negative value for the torque. A negative value of the external torque corresponds to a flexing of the joint, leading to a corresponding rotation (with negative angular velocity) of the joint by the motor. A positive value of the external torque represents an extension of the joint, leading to a corresponding rotation (with positive angular velocity) of the joint. Accordingly, the STS movement can be simulated for both correct and incorrect applications of the skill by the nurse, as stated in the requirement. If the robot patient is prompted to bend down by the nurse, the robot will voluntarily flex the knee and hip joints first. If the robot is then lifted, it will extend the hip and knee joints by itself. Conversely, if the robot patient is not prompted to bend down by the nurse during the STS transfer, the robot will extend the hip and knee joints without flexion.

The motor commands generated by the motion planning are sent to the profile position control. In the profile position mode, a target position ($\Delta\theta$) and angular velocity ($w$) are applied to the trajectory generator, as shown in Figure 9. Without profile position control, positioning would proceed without an accompanying profile. However, with profile position control, a position demand internal value is generated for the position PID control loop. The trajectory generator also computes the position demand internal value depending on the configured profile velocity, acceleration, and deceleration.

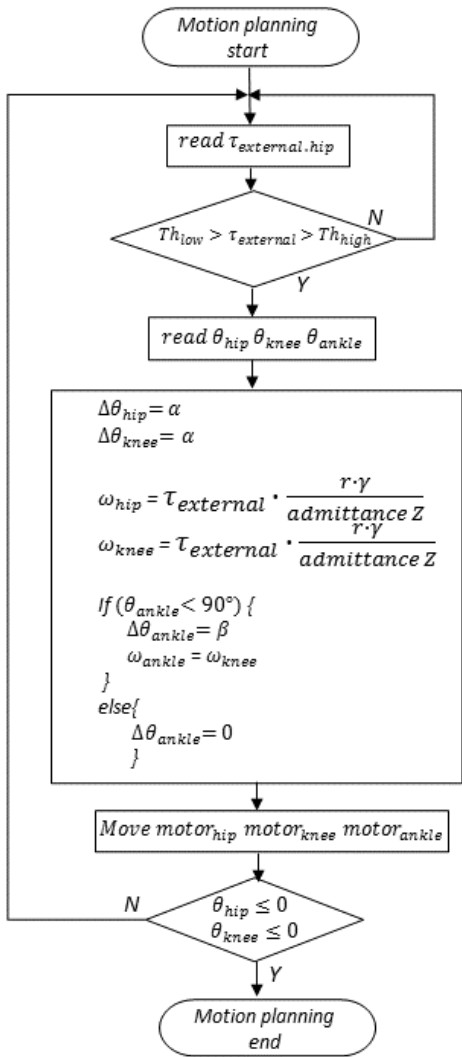

**Figure 8.** Motion planning of robot's lower limbs.

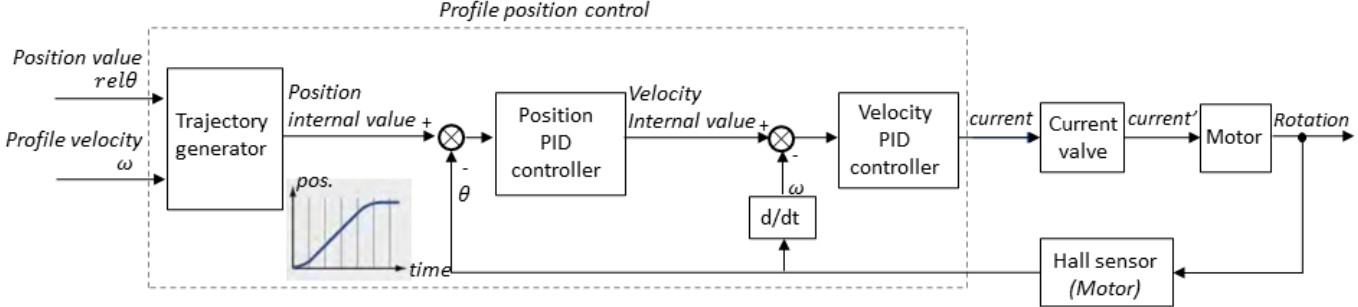

**Figure 9.** Block diagram of profile position control.

## 4. Experiment

### 4.1. Purpose

The purpose of the experiment was to verify whether the joint torque and external torque measured in the robot patient's lower limbs could distinguish between correct and incorrect application of transfer skill by introducing the proposed STS control method. We also intended to investigate and determine the most significant measurements among the joints of the lower limbs.

### 4.2. Participants and Procedure

Four nursing teachers with up to ten years of experience in nursing education were invited to participate in the experiment. The study was approved by the ethics review committees at the University of Tokyo and Tokyo Ariake University of Medical and Health Science. All participants were informed about the purpose of the study and provided written informed consent prior to the experiment.

The experimental procedure included a brief orientation on the robot and the experimental trials. At the outset, basic information about the robot, safety issues, and precautions was provided to the participants. During the experiment, the nursing teachers were asked to assist the patient in standing up, demonstrating both correct and incorrect methods, as show in Figure 10. The participants were asked to repeat each method three times. The correct method required the participants to squat down and lower their waist in preparation for helping the patient to stand up. The incorrect method required the participants to not bend their knees or lower their waist. All four participants were subjected to randomization and counterbalancing techniques.

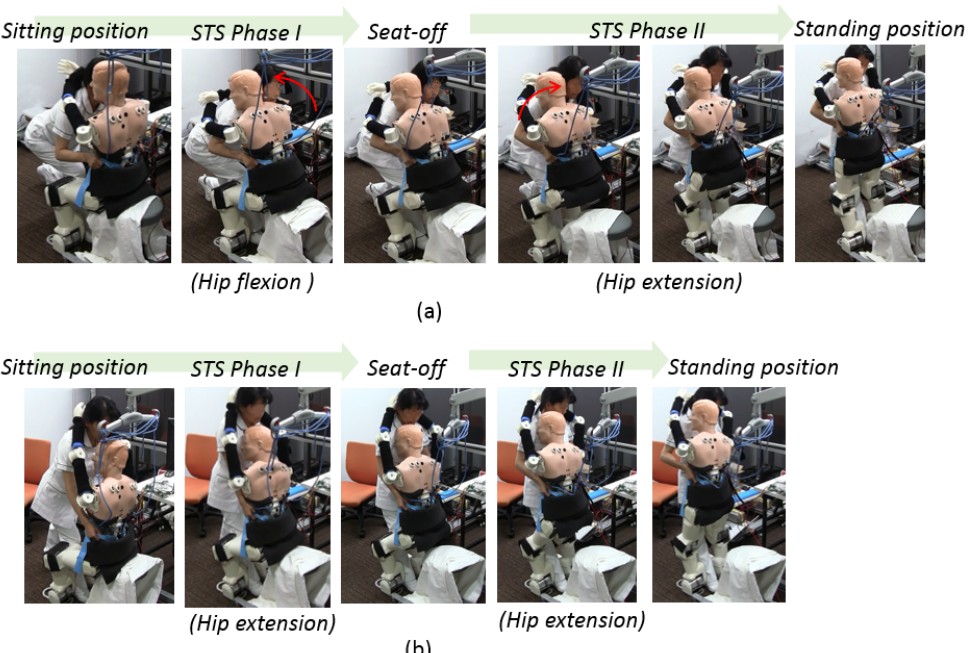

**Figure 10.** STS transfer skill performed by nursing teacher with (**a**) correct and (**b**) incorrect methods

### 4.3. Experimental Setting

The robot patient was initially sitting on a chair. For safety reasons, the chair was suspended on a patient lifter (KQ-781, PARAMOUNT BED CO., LTD.) to avoid accidental falling, which could injure the participants during the experiment. The height of the chair was adjusted to 60 cm, which allowed the robot's knee joint to bend to approximately 90°, allowing the robot patient to assume a sitting position. A power supply was connected to the robot with an emergency switch, which was held by a staff member during the experiment to prevent runaway. Soft materials were attached to the robot to avoid injury to the participants in the case of a collision. A camera was installed to record the experimental trials.

According to a previous study [53,54], the actual patient torque may be lower than that observed in healthy persons by approximately 20 to 40%. Therefore, a maximum torque of 28.52 Nm was set for the robot patient's lower limb joint. This value was lower than that required in the robot patient's knee joint when the robot was in a standing position, and meant that nursing assistance was required. The corresponding maximum current of the motor ($\pm$4000 mA) was set for the current valve to restrict the joint torque.

### 4.4. Analysis of the Two Phases of STS Movement

According to former studies, the raw data obtained from STS movement were in the form of a time series and usually separated into distinct phases for analysis and comparison. In this study, the two phases (I and II) were separated by the seat-off moment, which refers to the instant at which there is zero load on the seat after the hip joint has moved away from the chair [55]. Previous studies have shown that measurement of the movement of the lower limbs can reveal significant differences among the phases in the STS movement, most significantly between phases I and II. In this study, the obtained time series data were segmented into two phases, and the maximum, minimum, and average values in each phase were extracted to compare correct and incorrect methods.

### 4.5. Result

For the statistical analysis, a paired t-test was used to obtain the *p*-value. The significance level was set as 0.05, which indicates a significant difference between two samples. The results obtained in this study were analyzed with a paired t-test to determine whether a significant difference existed between the correct and incorrect methods of providing standing assistance. Furthermore, the maxima, minima, and averages in the two STS phases were extracted and calculated. All statistical analyses were based on the average of three trials on each participant.

#### 4.5.1. External Torque

The external torque measured at the hip, knee, and ankle joints are presented in Table 3, and the external torque measured at the hip joint is depicted in Figure 11. The yellow area represents phase I of the STS transfer. According to Figure 11a, when using the correct method of providing standing assistance, a negative external torque value was necessary to flex the hip joint in phase I of the STS transfer, whereas a positive external torque value was necessary to extend the hip joint in phase II of the STS transfer. In contrast, when using the incorrect method of providing standing assistance, a negative external torque value was barely observed in phase I of the STS transfer when the robot patient bent its hip and knee joints; an external torque was only measured while extending the hip joint, as illustrated in Figure 11b.

**Table 3.** External torque measured in the robot patient.

| Phase | Value | Correct | | | | Incorrect | | | | *p*-Value |
|---|---|---|---|---|---|---|---|---|---|---|
| | | Teacher A | Teacher B | Teacher C | Teacher D | Teacher A | Teacher B | Teacher C | Teacher D | |
| I | *min. hip joint external torque | -21.30 | −21.86 | −25.12 | −16.68 | −4.47 | −5.65 | −2.34 | −1.52 | 0.002 |
| | *min. knee joint external torque | −26.08 | −27.40 | −22.00 | −21.21 | −3.80 | −3.77 | −7.20 | −1.41 | 0.002 |
| | *min. ankle joint external torque | −30.07 | −22.87 | −20.85 | −39.89 | −2.17 | −2.08 | −1.10 | −4.02 | 0.006 |
| | *average hip joint external torque | −5.39 | −7.74 | −9.05 | 2.30 | 3.37 | 0.35 | 2.60 | 5.20 | 0.023 |
| | *average knee joint external torque | −6.94 | −6.94 | −6.22 | −0.82 | 0.69 | 2.39 | −0.82 | 1.02 | 0.033 |
| | *average ankle joint external torque | −6.55 | −5.12 | −5.31 | −0.30 | 2.08 | 2.40 | 2.64 | 0.24 | 0.047 |
| II | *max. hip joint external torque | 24.00 | 18.40 | 20.90 | 23.83 | 37.56 | 27.04 | 36.24 | 27.66 | 0.028 |
| | max. knee joint external torque | 20.37 | 21.58 | 20.00 | 36.43 | 17.22 | 27.18 | 28.67 | 40.01 | 0.238 |
| | *min. ankle joint external torque | −36.9 | −33.36 | −35.7 | −46.13 | −14.17 | −16.59 | −30.04 | −30.55 | 0.023 |
| | *average hip joint external torque | 5.14 | 3.51 | 2.42 | 8.49 | 11.49 | 6.62 | 9.73 | 10.47 | 0.034 |
| | average knee joint external torque | 2.12 | 4.01 | −0.12 | 31.87 | −4.32 | 5.26 | −4.99 | 16.02 | 0.164 |
| | *average ankle joint external torque | −30.24 | −24.13 | −19.96 | −35.54 | −3.04 | −7.03 | −12.04 | −11.00 | 0.021 |

* Represents a significant difference between correct and incorrect method.

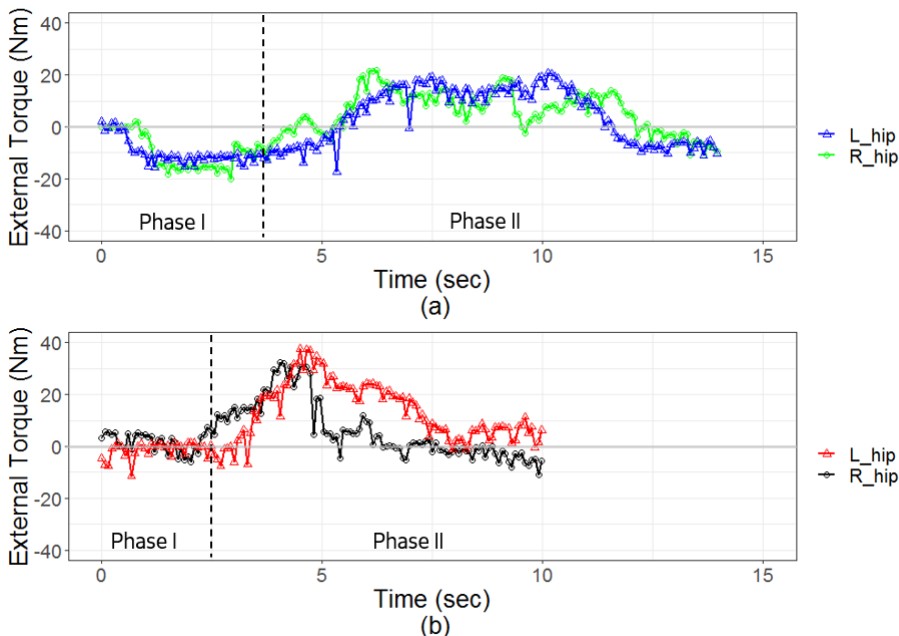

**Figure 11.** External torque measured from the hip joint using the correct (**a**) and incorrect (**b**) methods of standing assistance.

As shown in Figure 12, the external torque measured from the knee joint is similar to that measured from the hip joint. Using the correct method of standing assistance, the negative external torque value that was necessary to bend the knee joint was observed in phase I of the STS transfer. In phase II of the STS transfer, a positive external torque value necessary to extend the knee joint was observed, as shown in Figure 12a. The negative external torque value necessary to bend the knee joint was hardly observed in phase I of the STS transfer when using the incorrect method. However, the torque necessary to extend the knee joint was measured in both phases I and II of the STS transfer, as shown in Figure 12b.

The external torque measured from the ankle joint is shown in Figure 13a,b. Using the correct method, the negative external torque value that was necessary to bend the ankle joint was observed in phase I of the STS transfer, and the external torque necessary to flex the ankle joint was measured in phase II of the STS transfer, as shown in Figure 13a. Using the incorrect method of standing assistance, the negative external torque value that was necessary to bend the ankle joint was not observed in phase I of the STS transfer, whereas the torque necessary to flex the knee joint was measured in phase II of the STS transfer, as shown in Figure 13b.

According to the statistical analysis presented in Table 3, significant differences in the minimum external torque values required to flex the hip ($p = 0.002$), knee ($p = 0.002$), and ankle ($p = 0.006$) joints during phase I of the STS transfer were observed between the correct and incorrect methods of providing standing assistance. In addition, significant differences in the average external torque were observed in the hip ($p = 0.023$), knee ($p = 0.033$), and ankle ($p = 0.047$) joints. Furthermore, in phase II of the STS transfer, a significant difference was observed in the maximum external torque ($p = 0.028$) and the average external torque ($p = 0.034$) measured from the hip joint. A significant difference in the minimum external torque ($p = 0.023$) and the average external torque ($p = 0.021$) of the ankle joint was also observed in phase II of the STS transfer.

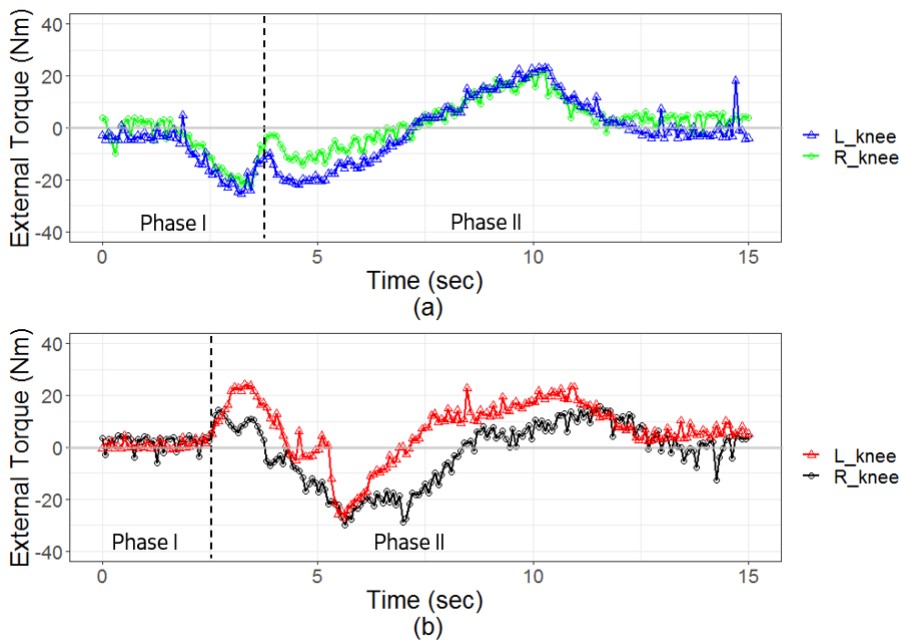

**Figure 12.** External torque measured from the knee joint using the correct (**a**) and incorrect (**b**) methods of standing assistance.

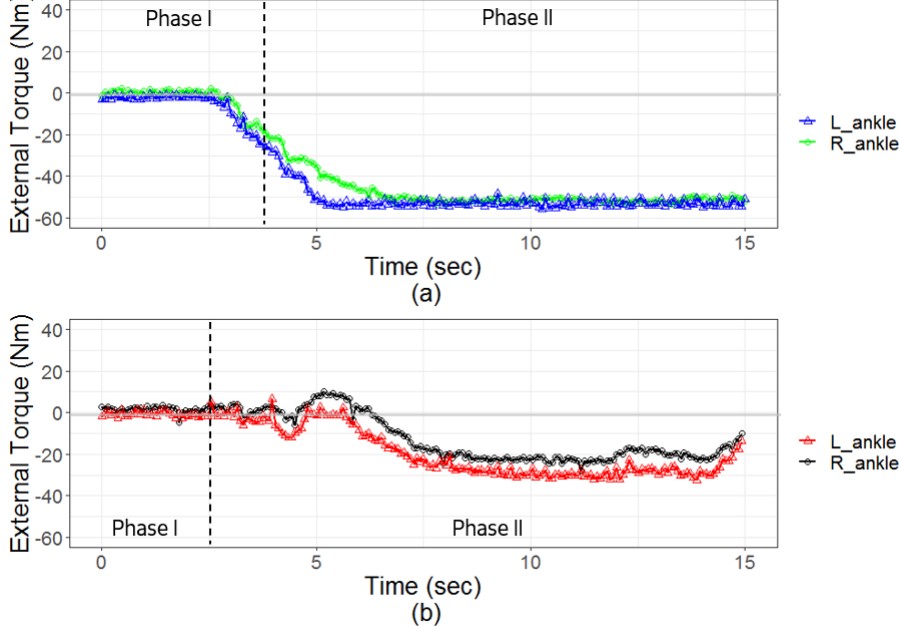

**Figure 13.** External torque measured from the ankle joint using the correct (**a**) and incorrect (**b**) methods of standing assistance.

### 4.5.2. Joint Torque

The joint torque of the robot patient's lower limbs was computed using the motor current. The joint torque values of the hip and knee joints are presented in Figures 14–16, and the results of the statistical analysis are presented in Table 4.

The joint torque measured from the hip joint is presented in Figure 14. When the correct method was applied, a negative flexing torque value was observed in phase I of the STS transfer, whereas an extending torque was observed in phase II of the STS transfer, as shown in Figure 14a. Conversely, using the incorrect method, only a minimal amount of flexing torque was observed in phase I of the STS transfer, whereas an extending torque was measured in both phases I and II of the STS transfer, as shown in Figure 14b.

**Table 4.** Joint torque measured in the robot patient.

| Phase | Value | Correct | | | | Incorrect | | | | *p*-Value |
|---|---|---|---|---|---|---|---|---|---|---|
| | | *Teacher A* | *Teacher B* | *Teacher C* | *Teacher D* | *Teacher A* | *Teacher B* | *Teacher C* | *Teacher D* | |
| I | *average hip joint torque | -8.58 | -5.73 | −4.37 | −3.34 | 13.74 | 12.06 | 7.33 | 13.72 | 0.004 |
| | *average knee joint torque | −14.94 | −10.86 | −7.74 | −9.89 | 16.94 | 11.96 | 2.26 | 11.74 | 0.017 |
| | *average ankle joint torque | −10.08 | −7.60 | −6.71 | −5.25 | −0.005 | −0.001 | −0.003 | −0.003 | 0.005 |
| II | *average hip joint torque | 15.49 | 16.94 | 17.94 | 14.17 | 17.70 | 21.94 | 19.94 | 20.19 | 0.032 |
| | *average knee joint torque | 12.62 | 9.43 | 11.99 | 11.38 | 15.62 | 15.00 | 13.25 | 16.20 | 0.032 |
| | *average ankle joint torque | 1.64 | 1.37 | 1.832 | 1.29 | −0.003 | −0.008 | 0.353 | −0.001 | <0.001 |

* Represents a significant difference between correct and incorrect method.

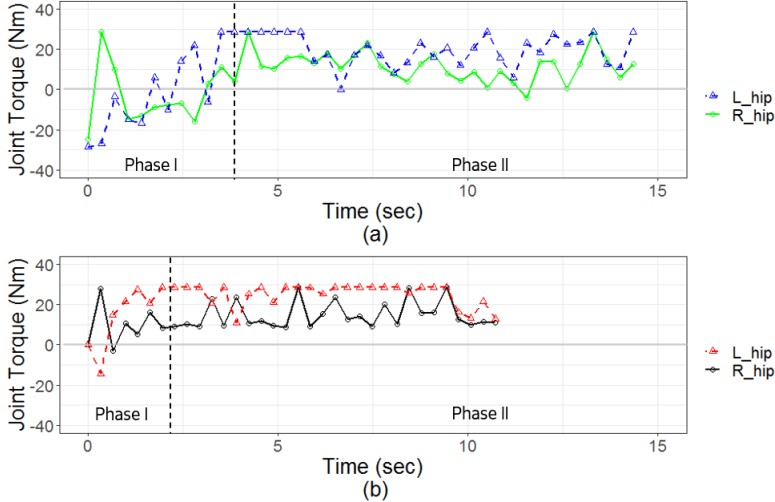

**Figure 14.** Joint torque measured from the hip joint using the correct (**a**) and incorrect (**b**) methods of standing assistance.

According to Figure 15a,b, the joint torque measured in the knee joint is similar to that measured in the hip joint. The negative joint torque value that was necessary to bend the knee joint was measured in phase I of the STS transfer; subsequently, the positive joint torque value that was necessary to extend the knee joint was observed in phase II of the STS transfer when using the correct method, as shown in Figure 15a. Conversely, the negative joint torque value that was necessary to bend the knee joint was barely observed in phase I of the STS transfer when using the incorrect method, whereas the joint torque that was necessary to extend the knee was observed in both phases I and II of the STS transfer, as can be seen in Figure 15b.

In addition, the torque of the ankle joint is shown in Figure 16a,b. When using the correct method, the motor exerted a torque to first flex the ankle joint in phase I of the STS transfer and then extend the ankle joint in phase II of the STS transfer. In contrast with the case for the correct method, almost no joint torque was necessary to flex the ankle joint in both phases I and II of the STS transfer when using the incorrect method, as shown in Figure 16b.

Based on the results obtained when using the correct method, the average joint torque values necessary to flex the joint during phase I of the STS transfer were negative. This observation reveals the dynamic status of the robot patient when flexing the hip, knee, and ankle joints during phase I of the STS transfer. Conversely, a positive value for the average joint torque was obtained when incorrectly applying the method during phase I of the STS transfer, which suggests that the robot patient exerts a voluntary force to extend the lower limbs when the method is applied incorrectly. Accordingly, a significant difference was

observed at the hip ($p = 0.004$), knee ($p = 0.017$), and ankle ($p = 0.005$) joints during phase I of the STS transfer when the method was applied correctly and incorrectly.

Furthermore, a significant difference was observed in the average joint torque at the hip ($p = 0.032$) and knee ($p = 0.032$) joints, which reveals a greater extending joint torque value in phase II of the STS transfer when the method is incorrectly applied. Moreover, a significant difference was obtained in the average torque of the ankle joint ($p < 0.001$), which reveals an extending joint torque in phase II of the STS transfer when the method is correctly applied. Almost no joint torque occurred in the ankle joint in phase II of the STS transfer.

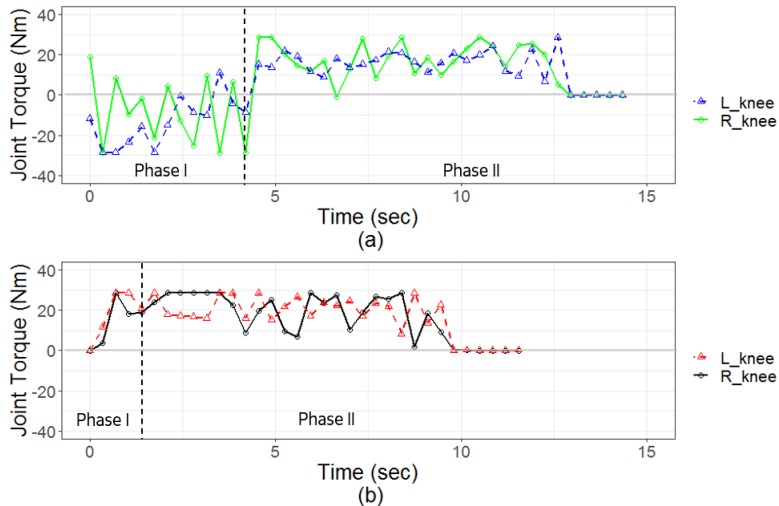

**Figure 15.** Joint torque measured from the knee joint using the correct (**a**) and incorrect (**b**) methods of standing assistance.

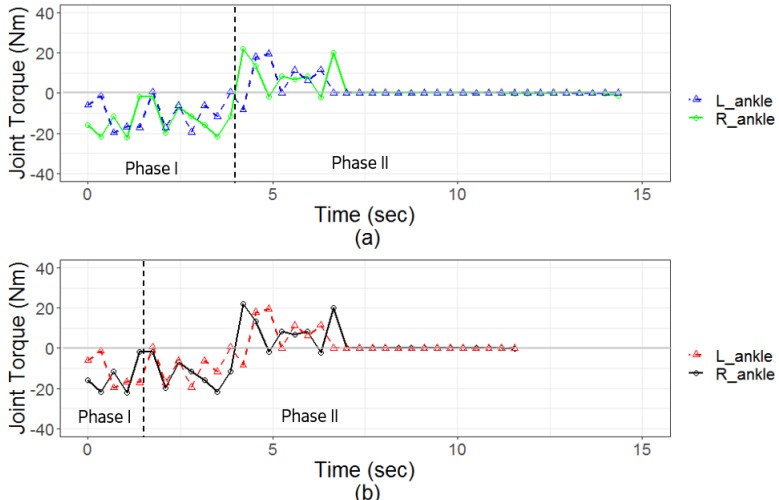

**Figure 16.** Joint torque measured from the ankle joint using the correct (**a**) and incorrect (**b**) methods of standing assistance.

## 5. Discussion

According to the results, statistically significant differences were observed in most of the joint torque and external torque values when compared with the values observed using the incorrect method. The joint torque represents the status of the robot's voluntary movements during the STS transfer, whereas the external torque reveals the condition of the nurse's support during the STS transfer. The results supported the hypothesis raised at the start of the study, which posited that both the external torque and the joint torque are crucial factors and can exhibit differences during STS transfer. Significant differences

of this nature are enabled by developing the robot's lower limbs with active joints and incorporating an STS control method that accommodates the actions of the patient as well as the nurse.

### 5.1. External Torque

Using the correct method (i.e., lowering the waist and squatting down to assist the patient to stand), a minimum negative external torque value to bend the trunk was observed in phase I of the STS transfer. This torque was not observed when using the incorrect method. This difference may be explained as follows—by lowering the body, the nurse is able to support the patient in standing up by exerting an upward force and a forward force on the patient. In contrast, when using the incorrect method, an upward force is still applied, but the nurse finds it difficult to exert a forward force to bend the patient's trunk.

In addition, when using the incorrect method, a greater maximum external torque was observed at the hip joint during phase II of the STS transfer. This observation revealed that the incorrect method might lead to a greater external torque affecting the dynamic status of the patient; thus, the movement of the patient may become unstable when using the incorrect method. When using the correct method, the patient would not have such a drastic external torque exerted by the nurse, which would ensure the safety and comfort of the patient during the STS transfer. In addition, when using the incorrect method, the average external force exhibited a significant difference when compared with the correct method, which revealed a greater extending (positive) torque value during phase II of the STS transfer. This possibly occurs because the duration of STS transfer is shorter when using the incorrect method, and the nurse exerts a greater force to lift the patient into a vertical position.

A significant difference was observed in the minimum and average external torque values measured at the ankle joint in phase II of the STS transfer, and both values were smaller when using the correct method. This suggests that a greater degree of external torque is necessary to bend the ankle joint when using the correct method. This is plausible, as in the correct method, the patient's trunk was bent downward first before the trunk and the lower limbs were leaned forward, resulting in a torque that flexed the ankle joint. In contrast, when applying the incorrect method, the patient was lifted to a vertical position; therefore, the ankle joint was positioned at an angle of approximately 90°, and almost no negative external torque was required to bend the ankle joint.

The maximum and average external torque values measured from the knee joint were not significantly different for the correct and incorrect methods of standing assistance. This observation can be explained as follows: because the nurse grasps the waistband to assist the robot patient in standing, sudden changes in the torque can be sensitively measured from the hip joint, but may not be measurable from the knee joint, as it is very far from the point of application.

### 5.2. Joint Torque

The average joint torque values in each of the hip, knee, and ankle joints were significantly different when comparing the correct and incorrect methods of providing standing assistance in both phases I and II of the STS transfer. During phase I of the STS transfer, the average joint torque was negative, which represents the joint torque necessary to flex the joint when the nurse uses the correct method of providing standing assistance. Conversely, when the incorrect method of providing standing assistance was used, the average joint torque was positive at the hip and knee joints, which represents the joint torque necessary to extend those joints. These observations occurred because the motor's velocity was proportional to the external torque, as proposed in the control method; thus, the motor torque increased when the external torque increased. In addition, the rotational direction in the hip and knee joints was based on the direction of the external torque when measured at the hip joint. During phase I of the STS transfer, the external torque necessary to bend the trunk was measured while the correct method was being applied, whereas the external

torque necessary to extend the trunk was measured when the incorrect method was being applied. Such results resulted in the different directions (i.e., flexion, extension) of the joint torque at the hip and knee joints when using the correct and incorrect methods of providing standing assistance.

Furthermore, in phase II of the STS transfer, the torques at the robot patient's hip and knee joints differed significantly. The degree to which a particular joint could be extended by the application of torque varied according to whether the correct or incorrect method of providing standing assistance was being used. This variation occurred because the robot patient exerted a particular joint torque in accordance with the amount of external torque measured at the robot patient's hip joint. Therefore, when a significant difference in the average external torque was observed, a significant difference in the joint torque at the hip and knee joints was also observed. This suggests that a greater joint torque was required for the robot patient when the incorrect method was applied during phase II of the STS transfer.

These results are consistent with those reported in previous studies. For example, in [56], a significant difference was observed in the joint torque during STS transfer when comparing the results for disabled subjects against those for healthy subjects. In addition, it has been suggested that the STS movement of elderly patients is the result of poor application of force and the slowing of effort [57]. These results are consistent with the control method used in the robot patient in this study. To simulate an elderly person or a patient with severe disability, we set the lower current limit of the motor to an appropriate value. Thus, the force exerted by the robot patient was decreased, which allowed the nurses to obtain clinical experience by practicing the correct method for providing standing assistance to the robot patient.

## 6. Conclusions and Future Work

In accordance with the ultimate objective of employing a robot patient in the training of STS transfer skills and assessing the performance of these skills, the joint torque and external torque were measured at the robot's lower limbs. In addition, a control method was proposed to reproduce the patient's STS movement depending on the nurse's skill. To achieve such requirements, this study adopted admittance control of motor movement by an external force. Both the direction and quantity of the external torque measured at the hip joint served as the variables to control the robot's standing movement. The robot's lower limbs were redesigned from passive and free joints to a full complement of active joints with compliant units, which allowed the external force to be measured and the stiffness of a human joint to be reproduced.

According to the results obtained in this study, the minimum and average values of the joint torque and the average value of the external torque, measured during phase I of the STS transfer, revealed a significant difference when the incorrect method was used. The maximum and average joint torque values and the external torque in the hip joint, measured during phase II of the STS transfer, increased when the incorrect method was used. However, the maximum external torque in the knee joint did not reveal a significant difference because the nurse provided support at the waistband, which is anatomically positioned very far from the knees. Moreover, the average joint torque can reflect the dynamic status of the robot patient. However, data cannot be used to assess the skill applied by the nurse when the current limitation of the motor is lowered, as the difference in measurements may fail to be significant.

In the future, a larger variety of incorrect transfer methods will be included in the system, and the threshold of each step will be determined by employing a larger number of participants. Furthermore, the learning effectiveness will be measured by comparing the pre- and post-test measurements, which will allow examination of improvements in applying skills gained through practice with the robot patient. Finally, applications for practice with robot patients are expected to extend to other patient handling skills.

**Author Contributions:** C.L., T.O. and J.O. conceived and designed the study; the robot development and data analysis were done by C.L.; also M.K.-P., J.M., Y.K. and M.N. contributed the nursing acknowledge performed the experiments; Z.Z. assisted the experiment and the controlling of robot; N.K. participated into the discussion. All authors have read and agreed to the published version of the manuscript.

**Funding:** This work was partially supported by JSPS KAKENHI Grant Number 20H04261, 19H05730, and 19H04154.

**Institutional Review Board Statement:** All procedures were approved by the Ethics Committee at the University of Tokyo, and all participants provided their written informed consent prior to enrollment

**Informed Consent Statement:** Informed consent was obtained from all subjects involved in the study. Also written informed consent has been obtained from the patients to publish this paper.

**Data Availability Statement:** The data used to support the findings of this study are available from the corresponding author upon request.

**Acknowledgments:** For the development of the robot patient, the hardware of reduction gear was supported by TPI Co., Ltd.

**Conflicts of Interest:** The authors declare no conflict of interest. Also the funders had no role in the design of the study; in the collection, analyses, or interpretation of data; in the writing of the manuscript, or in the decision to publish the results.

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
