# Peer review of "Development of Robot Patient Lower Limbs to Reproduce the Sit-to-Stand Movement with Correct and Incorrect Applications of Transfer Skills by Nurses"

_applsci, doi:10.3390/app11062872_

Round 1

Reviewer 1 Report

The paper entitled “Development of Robot Patient Lower Limbs to Reproduce the

Sit-to-Stand Movement with Correct and Incorrect Applications

of Transfer Skills by Nurses” is well written and if fits in the scope of the MDPI applied sciences Journal. The references are adequate and the new contributions are adequate. Some aspects can be improved, such as to restructure section 2 because its dimension is not adequate. The sections dimension must be balanced, so in this sense section 2 must be expanded or merged with another section. Figures 8, 10, 11, 12, 13, 14, 15 and 16 can have a significant quality increase. The authors must avoid to refer to themselves in the first person ‘we hope´ ‘we measure’.

The presented approach has some novelty and it has a huge impact in the quality of life of the health care workers and in the life of the elders that need their assistance. The existence of tools to improve the skills of the health care workers in a controlled environment is very important, so in this context new contributions to this field are very relevant.

Author Response

Please refer to the attached PDF file. 

Reviewer 2 Report

Manuscript ID: applsci-1129021

Title: Development of Robot Patient Lower Limbs to Reproduce the Sit-to-Stand Movement with Correct and Incorrect Applications of Transfer Skills by Nurses

Authors: Chingszu Lin, Taiki Ogata, Zhihang Zhong, Masako Kanai-Pak, Jukai Maeda, Yasuko Kitajima, Mitsuhiro Nakamura, Noriaki Kuwahara, Jun Ota Submitted to section: Robotics and Automation,

The authors describe their work on developing a robot that mimics an elderly person with reduced ability to perform a sit to stand movement by itself. The robot is used to teach nurses the correct methods of standing assistance. Therefore torque measuring devices are placed in the robot such that the external applied toque due to the standing assistance, as well as the torque of the joints of the robot can be measured. By experiments with four nurse teachers performing the correct and incorrect methods of standing assistance at the robot, significant differences are observed that can help in teaching the novices in the correct assistance.

The paper is very well written. However, while reading I came up with a few comments and questions about the manuscript, which are discussed below.

Questions:

Q1:     Table 1 lists the important values of the robot patient. I am wondering why the range of movement of the hip joint starts at -30°. From Figure 1 a can’t see that the upper body leans backwards and at line 260 ff. the range of the hip joint motion is also only mentioned between 0° and 120°. Can you please specify the -30°.

Q2:     I haven’t found a precise description of p-value in the manuscript. Please make clear what that value represents and how you obtain it from measurements.

Q3:     In all diagrams you show the torque of the left and right joints. Can you explain why you need to distinguish between both when the motion of the robot is symmetrical? I assume that drawing the graphs of correct and incorrect method of assistance in one diagram will improve the comparison of the differences

Comments:

C1:      The formula symbols in equation (1) are not inserted again with the descriptions in the text, which can lead to misunderstandings. Please explain all formula symbols in the text.

C2:      The tables look very disorganized. The insertion of lines and the increase of the column width would lead to a better readability. In particular, table 3 could be made clearer by, for example, specifying torque in phase I or torque in phase II only once for all rows and sorting the values not by teacher but by correct and incorrect methods of standing assistance.

C4:      The quality of Figure 8 is poor. Can you increase the quality?

C5:      The diagrams of Fig. 10 and Fig 11 have not the same width, which makes it hard to compare. Please try to unify the width of the diagrams. Probably the highlighting of the zero line further promotes the differences of the two diagrams.

Typos:

line 22:           As the world’s population is aging[1],…

                        Insert a space in front of the reference

line 666:        phase 1 should be replaced by phase I

Author Response

Please refer to the attached PDF file. 
